# Natural and Engineered Nanomaterials for the Identification of Heavy Metal Ions—A Review

**DOI:** 10.3390/nano12152665

**Published:** 2022-08-03

**Authors:** Joseph Merillyn Vonnie, Bong Jing Ting, Kobun Rovina, Nasir Md Nur’ Aqilah, Koh Wee Yin, Nurul Huda

**Affiliations:** Faculty of Food Science and Nutrition, Universiti Malaysia Sabah, Kota Kinabalu 88400, Malaysia; vonnie.merillyn@gmail.com (J.M.V.); applebongjingting@gmail.com (B.J.T.); aqilah98nash@gmail.com (N.M.N.A.); weeyin@ums.edu.my (K.W.Y.); drnurulhuda@ums.edu.my (N.H.)

**Keywords:** heavy metal ions, sensor, nanotechnology, green synthesis, environmental-friendly, sustainable

## Abstract

In recent years, there has been much interest in developing advanced and innovative approaches for sensing applications in various fields, including agriculture and environmental remediation. The development of novel sensors for detecting heavy metals using nanomaterials has emerged as a rapidly developing research area due to its high availability and sustainability. This review emphasized the naturally derived and engineered nanomaterials that have the potential to be applied as sensing reagents to interact with metal ions or as reducing and stabilizing agents to synthesize metallic nanoparticles for the detection of heavy metal ions. This review also focused on the recent advancement of nanotechnology-based detection methods using naturally derived and engineered materials, with a summary of their sensitivity and selectivity towards heavy metals. This review paper covers the pros and cons of sensing applications with recent research published from 2015 to 2022.

## 1. Introduction

Heavy metal pollution has become a concern for global sustainability and has impacted various environmental components, including terrestrial and aquatic ecosystems. They are one of the micropollutants with an increased environmental problem due to rapid advancement in industrial and mining activities over the last few decades. Heavy metal contamination is mainly due to the chemicals generated from industrial or household wastes, burning of fossil fuels, use of fertilizers, cosmetics, and their by-products [1]. It is non-biodegradable, water-soluble, has an extended half-life, and accumulates in food chains and living organisms. As a result, releasing them extensively can severely damage the environment and promote the development of many dangerous diseases that pose a significant public health concern [2]. Among eclectic heavy metals, lead (Pb), cadmium (Cd), mercury (Hg), chromium (Cr), and arsenic (As) are highly toxic as they can lead to severe concerns to the environment and human health [3].

Therefore, monitoring heavy metals in the environment, drinking water, and agricultural products is essential. The conventional detection methods used for the identification of traces of heavy metals include atomic absorption/emission spectrometry (AAS/AES), hybrid generation atomic absorption spectrometry (HG-AAS), inductively coupled plasma optical emission spectrometry (ICP-OES), inductively coupled plasma mass spectrometry (ICP-MS) and high-performance liquid chromatography (HPLC) [4]. Although these methods provide benefits including high sensitivity and accuracy, they also suffer from some drawbacks such as being costly, time-consuming, requiring expert personnel and complex equipment, multi-step procedures, and being labor intensive, making them non-practical for on-site analysis. Hence, developing a simple, cheap, and rapid heavy metal detection method highly demands environmental safety.

In the past decade, nanomaterials have evolved into a significant class of materials that includes examples with at least one dimension ranging from 1 to 100 nm [5]. It is classified into four-dimensional nanostructures in single, fusion, and aggregation forms or agglomeration of tubular, irregular, and sphere shapes. Common nanomaterials employed for heavy metal identification are carbon dots [6], graphene [7], gold/silver/copper nanoparticles [2], and metal nanoclusters [8]. Natural and engineered nanotechnology-based detection methods act as alternatives to conventional laboratory methodologies. This is because the nano-sized particles with a large surface area for efficient interaction with ions could improve the performance of sensors or detectors in terms of sensitivity, the limit of detection, selectivity, and reproducibility towards specific contaminants in the environment [9]. Nowadays, researchers are inspired to develop novel nanotechnology-based sensors for heavy metal detection using readily available natural resources. Using natural resources as essential materials in developing sensing platforms showed the ease of synthesis and functionalization, low cost, environmental-friendly nature, high availability, and sustainability. Besides, metallic nanoparticles’ green synthesis or biosynthesis is highly acceptable compared with physical and chemical processes. The nanoparticles derived from natural resources such as plants and microorganisms are considered clean, eco-friendly, cost-effective, non-toxic, and safe for environmental applications [10].

Despite the increasing development of eco-friendly and sustainable approaches in recent years, limited articles still review the nanotechnology-based detection method using naturally derived materials and highlight its sensing applications in monitoring heavy metals. Therefore, this review emphasizes the natural and engineered nanomaterials that can be applied as sensing reagents to interact with metal ions (Figure 1) or reducing and stabilizing agents to synthesize metallic nanoparticles for identifying heavy metals. The synthesis method and detection techniques of each potential natural resource for heavy metal detection are also highlighted and concluded with a brief discussion regarding their pros and cons for sensing applications. In addition, this review focuses on the recent advancement of nanotechnology-based detection methods using natural and engineered nanomaterials with a summary of their sensitivity and selectivity towards heavy metals.

## 2. Application of Natural-Derived

Natural resources containing high carbon content are potential and suitable carbon-driven precursors for synthesizing carbon-based nanomaterials as sensing reagents for identifying heavy metals. The carbon source obtained from food and industrial wastes has been given particular attention to be properly used to develop sensing platforms in environmental applications, increasing by-product value, decreasing environmental pollution, and minimizing energy consumption. Natural resources such as curcumin, eggshells, and spinach extracts are used as heavy metal adsorbent materials. Metal collaborative approach or coordination with the introduction of beneficial collections in protein, lipids, and sugars on cell dividers may be rural side effects of metal particle adsorption [11]. Recently, much effort has been put into developing the green synthesis of carbon dots (CDs) from natural resources for sensing applications. CDs are nanoparticles of nanometer size below 10 nm that exhibit good properties, including low toxicity, water solubility, high chemical stability, high resistance to photo-bleaching, flexibility to surface modification, and good biocompatibility compared with the other organic dyes and heavy metal-based quantum dots [12]. The CDs can be prepared from various natural resources through simple synthesis techniques, including electrochemical oxidation, combustion/thermal, chemical change, microwave heating, arc-discharge, and laser ablation methods reviewed in detail by Das et al. [13]. However, the practical large-scale applications of the CDs synthesized from these cheap and abundant precursors are limited due to their relatively low quantum yield [14]. CDs also suffer from some drawbacks such as complexity of procedure, relatively large amounts of strong acids, and costly surface passivating agents being required to improve their fluorescence property and selective sensing of metals when applied in sensing applications [15]. 

### 2.1. Lotus Root

Lotus root (LR) is a common and inexpensive crop cultivated in China, Japan, India, and Australia. Generally, it consists of amino acids, alkaloids, glycoproteins, and polysaccharides, all of which contribute significantly to the production of CDs by serving as a plentiful source of nitrogen and carbon [16]. Gu et al. [14] synthesized fluorescent nitrogen-doped CDs using a one-pot microwave treatment. The treatment of LR to produce fluorescent nitrogen-doped CDs was free from any other surface passivation agents. Among a variety of metal ions, only Hg^2+^ ions induce noticeable quenching effects because of the greater affinity and faster chelating kinetics of Hg^2+^ ions towards the carboxylic, hydroxyl, and amino groups on the LR CDs surface than other metal ions. The results also showed that the LR CDs were highly sensitive toward Hg^2+^ ions with a linear range from 0.1 to 60.0 µm and a detection limit of 18.7 µm. Compared with the CDs synthesized from other natural resources, the LR CDs showed a higher quantum yield capable of emitting strong fluorescence at low concentrations. The unique properties of CDs derived from LR made it have great potential to be applied as sensing material in aqueous samples. However, they suffered from certain limitations, such as complicated, time-consuming procedures that may lead to chemical hazards.

### 2.2. Rose-Heart Radish

Rose-heart radish is a watermelon radish well known for its health benefits due to its water-soluble pigment, anthocyanin, which protects against cardiovascular diseases. The CDs fabricated from rose-heart radish could easily be functionalized due to the abundance of carbon, nitrogen, and oxygen elements from crude protein, amino acid, carbohydrates, and vitamins found in rose-heart radish. The high availability and low cost of rose-heart radish make it applicable for the large-scale fabrication of CDs. Based on the studies by Liu et al. [17], the fluorescent nitrogen-doped CDs with well-distributed size could be synthesized from rose-heart radish via one-pot hydrothermal treatment at 180 °C for 3 h. The developed nitrogen-doped CDs showed a relatively small distribution from 1.2 to 6.0 nm, a high fluorescent quantum yield of 13.6%, high biocompatibility, low toxicity, and good chemical stability. The prepared nitrogen-doped CDs are functional groups containing oxygen and nitrogen elements, contributing to good water solubility and easing further modifications and applications when used as fluorescent materials. Furthermore, Liu et al. [17] also explored that the nitrogen-doped CDs strongly responded to Fe^3+^ ions and gave rise to fluorescence quenching. Thus, it can be utilized as a novel sensing probe for the sensitive detection of Fe^3+^ ions with a linear range from 0.02 µm to 40 µm and a detection limit of 0.13 µm. The authors also explored that the fluorescence intensity of the CDs was improved with the increase of pH values from 2.0 to 7.0. In contrast, fluorescent behavior increased dramatically when pH was higher than 8.0.

### 2.3. Red Lentils

Red lentils are one of the widely consumed edible pulses and present great potential to be used as naturally occurring carbon and nitrogen sources for synthesizing carbon-based nanomaterials due to their high protein content, fats, and carbohydrates. Khan et al. [18] developed a green synthesis of water-soluble nitrogen-doped CQDs from red lentils via a one-step hydrothermal method. The prepared nitrogen-doped CQDs showed a quantum yield of 13.2% and exhibited excitation-dependent emission properties, and amorphous and graphitic nature. The nitrogen-doped CQDs showed a light yellow color under normal visible light, whereas a bright blue luminescence was observed under a UV lamp of the wavelength 365 nm. The nitrogen-doped CQDs derived from red lentils showed a high response to Fe^3+^ ions that can lead to fluorescence quenching, whereas no fluorescence intensity changes were observed with addition of other metal ions. The formation of complexes may explain the fluorescence quenching mechanism by binding Fe^3+^ ions to the hydroxyl group found on CQDs’ surface. The electron is then transferred from excited states of CQDs to half-filled 3d orbitals of Fe^3+^ ions, leading to non-radiative electron-hole pair recombination, which results in fluorescence quenching. This method showed the detection limit at 0.10 µm concentration of Fe3+ ions over the linear range of 2–20 µm with a correlation coefficient of R^2^ = 0.993. The nitrogen-doped CQDs derived from red lentils also showed stability in high salt conditions.

### 2.4. Coconut Coir

Coconut coir is the fibrous material found between the hard, internal shell and the outer coat of coconut fruit, consisting of about 75% fiber as its major constituent [19]. The coconut coir wastes can be utilized as carbon precursors for the fabrication of CQDs due to the high content of cellulose, hemicelluloses, and lignin. Chauhan et al. [20] reported a hydrothermal-based method using coconut coir as a precursor to synthesize the CQDs. The prepared coconut coir CDs showed a high quantum yield of 48.0%, a high solubility rate in aqueous media, and high stability over a wide pH range. The synthesized CDs also had a size of less than 10 nm and were monodispersed and spherical. The developed C-dots displayed a turn-on sensor for Cd^2+^ ions and a turn-off sensor for Cu^2+^ ion ions because of the enhancement and quenching of fluorescence in the emission intensity of CDs. For the Cd^2+^ ions, the enhancement of the emission peak of CDs was possibly due to the chemical interaction between the localized surface plasmon resonance at the surface of CDs and Cd^2+^ ions. The synthesized CDs showed the detection limits of 0.18 and 0.28 nm for Cd^2+^ and Cu^2+^ ions, respectively, with a regression coefficient reported as R^2^ = 0.99. The practical utilities of this method have also been validated over different water resources, including deionized water, tap water, sewage water, and groundwater. The developed sensor had the potential to be applied as a sensory probe for wastewater remediation because it was easy to use, rapid, cost-effective, and low-toxicity.

Prawn, squilla, krill, crab, lobster shells, and other marine crustacean wastes are industrial by-products that can cause serious environmental problems due to their resistance to biodegradation, insolubility in water, and high cost of waste processing [21]. The crustacean wastes contain a large quantity of chitin and its deacetylated derived components, such as chitosan, which can be used for detecting heavy metal residues in foods because they have a good chelating ability with heavy metals [15,22]. Therefore, crustacean waste can be properly used as a new platform for food safety and quality areas. Gedda et al. [15] demonstrated a green synthesis of fluorescent CDs from prawn shells via hydrothermal. The results showed that these developed CDs had an average diameter of 4 nm, allowing them to possess many excellent features, such as eliminating blue fluorescence under ultraviolet light (λ = 365 nm). Based on the studies, the CDs synthesized from prawn shells also had a high quantum yield at 9.0%, good stability, high monodispersity, and high water solubility. The prawn shell CDs were also found as effective sensing probes for detecting Cu^2+^ ions due to their excellent selectivity and sensitivity toward Cu^2+^ ions with a low detection limit of 5 nm. The fluorescence quenching of carbon dots by copper ions may be explained by the combination of Cu^2+^ ions with amine groups at the surface of CDs and its inner filter effect, leading to the formation of cupric amine complexes. Besides, the sensing ability of this novel detection method toward Cu^2+^ ions was also demonstrated through seawater sample analysis. The results showed the advantages of prawn shell CDs such as rapidity, simplicity, economical, sensitivity, and selectivity towards Cu^2+^ ions. Thus it was beneficial to be used as a sensing probe to monitor the level of Cu^2+^ ions in drinking water, river water, and seawater. In China, the Gardenia fruit was used as a natural yellow dye and has become a popular traditional Chinese medicine since its biological properties were discovered a few decades ago [23]. However, Sun et al. [24] used gardenia fruit as a precursor to prepare CDs using hydrothermal methods. The proposed nitrogen and sulfur co-doped CDs have an absolute quantum yield of up to 10.7%. The CDs can be quenched by Hg^2+^ when the sensing mode is turned off and display a good sensitivity and selectivity in detecting Hg^2+^ with a detection limit of 320 nm. A strong interaction between Hg2+ and the nitrogen and sulfur co-doped CDs could be responsible for the selectivity toward Hg^2+^.

### 2.5. Biomass Waste

Biomass-derived carbon materials show promising applications in water treatment, including removing heavy metal ions [25]. Since it is carbon-rich, low-cost, easy to access, ubiquitous, renewable, and environmentally friendly, biomass waste is an ideal precursor for carbon-based materials [26]. Utilizing waste-derived CDs can be economical and environmentally friendly while improving waste management [27]. In northern North America, Kentucky bluegrass is one of the most popular and widely propagated turfgrass species [28]. A facile hydrothermal method was recently used by Krishnaiah et al. [29] to synthesize nitrogen-doped carbon dots (NCDs) from the waste biomass of Kentucky bluegrass (KB). Using the hydrophilic optical properties of KBNCDs synthesized in this study, Fe^3+^ and Mn^2+^ ions were detected in an aqueous medium with good selectivity and sensitivity. LOD values for Fe^3+^ or Mn^2+^ ions were calculated as 1.4 µm and 1.2 µm, respectively, with detection ranges between 5.0 to 25 µm. Furthermore, a simple and green method for constructing a CD-based dual-mode fluorescent sensor from the waste biomass of wintersweet flowers (FW-CDs) was first proposed by Xia et al. [30]. FW-CD fluorescent probes were highly sensitive to Cr(VI) and Fe^3+^ and linear over a wide range of 0.1 to 60 µM and 0.05 to 100 µM, respectively. They also had low detection limits of 0.07 µM and 0.15 µM, respectively.

Carbon nano-onions are one of the carbon nanostructures with unique physicochemical properties due to the pronounced edge effects [31]. They are composed of concentric nanographene shells around a solid or hollow inner core that can be morphologically found between multiwall and fullerenes nanotubes [32]. Generally, due to their unique, peculiar onion-like structure, carbon nano-onions have physicochemical properties such as mechanical, optical, and electrochemical properties that make them suitable for sensing, bioimaging, drug delivery, and environmental remediation [33].

### 2.6. Flaxseed Oil

Flaxseed oil has a high polyunsaturated fatty acid content, especially omega-3 fatty acids [34]. It is widely used in nutraceuticals and functional foods due to its structural characteristics. Tripathi et al. [35] demonstrated the synthesis of onion-like carbon nanoparticles using flaxseed oil as the carbon precursor via traditional pyrolysis. The synthesized carbon nano-onions with a particular size of 4–8 nm became hydrophilic and stable in the aqueous phase due to the introduction of carboxyl functionalities via oxidative treatment. Due to the strong photoluminescence and large surface oxygenous functionalities, the developed water-soluble carbon nano-onions could be used as a fluorescent probe for sensing Al^3+^ ions at the detection limit of 0.77 µm of Al^3+^ ions without interference from other metal ions based upon the fluorescence turn-off technique. The binding of Al^3+^ ions with surface functional groups of carbon nano-onions led to quenching green photoluminescence emissions.

### 2.7. Curcumin

Curcumin, also known as diferuloylmethane, is the main natural polyphenol extracted from the rhizome of *Curcuma longa* (turmeric) and in others *Curcuma* spp. [36]. Recently, curcumin has received increasing attention by researchers because of its physiological properties, including anti-inflammatory, anti-diabetic, antitumor effects, treatment of skin burst and protection of the skin against UV and gamma radiation damage, immune system improvement, and wound healing [37]. Curcumin is widely used as a spice, food preservative, food flavoring, and food coloring. Curcumin coexists in two tautomeric forms, keto, and enol, depending on the solution pH [38]. In acidic and neutral conditions or a solid phase, curcumin acts as a hydrogen donor as the predominant keto form. In contrast, the enolic form of curcumin predominates under alkaline conditions to act as a metal ion chelator [39]. Curcumin that bears 1,3-diketones in tautomeric forms can chelate with various metal cations and anions to form complexes, leading to a color change from yellow to orange or red and fluorescence intensity changes [40].

Sheikhzadeh et al. [41] recently designed a novel colorimetric chemosensor based on curcumin extract embedded in bacterial cellulose nanofiber for selective detection of Pb^2+^ in rice. The presence of Pb^2+^ caused the color to change from orange to red, with detection limits of 9 µM and 0.9 µM through observation with the naked eye and image processing, respectively. Vonnie et al. [42] investigated the activity of a biofilm containing Aloe vera, green banana Saba, and curcumin in detecting Fe^2+^ ions. Using colorimetric analysis, good linearity (R^2^ = 0.9845) was found for Fe^2+^ ions concentrations of 0 to 100 ppm, with limits of detection and quantification found to be 27.84 ppm and 92.81 ppm, respectively. Zhang et al. [43] fabricated nanofibrous cellulose acetate/curcumin membranes by electrospinning to detect Pb^2+^ pollution rapidly. The authors also explained that the interaction between Pb^2+^ and curcumin in nanofibers caused a color change from bright yellow to light orange, which can be observed by naked eyes and a camera, indicating rapid detection of Pb^2+^ ions. The electrospun nanofibrous cellulose acetate/curcumin membranes with a thickness of 0.2 mm showed the detection limit at 1 mM of Pb^2+^. It was found that the fabricated membrane was selective toward Pb^2+^ ions at pH 9. Additionally, Raj and Shankaran [44] synthesized the curcumin-loaded cellulose acetate nanofibers that can be used as colorimetric sensor strips to detect Pb^2+^ ions. The strips sensor showed a higher selectivity towards Pb^2+^ ions at 1 mM concentration, with a detection limit of 20 µm at room temperature with a visual color change from yellow to orange when interacting with Pb^2+^ ions due to the formation of curcumin-Pb^2+^ ions complex. The linear correlation of this research was reported as R^2^ = 0.997, with a detection range of 0.12 ± 0.01 µm.

Earlier, Mohan and Prakash [45] fabricated eco-friendly hydrogel strips with curcumin and anthocyanin extracts from turmeric and red cabbage. These strips can be used for visual detection of Hg^2+^ and Cd^2+^ in river water samples under a UV chamber and emit color when in the presence of heavy metals (Hg^2+^ = green and Cd^2+^ = blue). The sensor’s response time is 5 min, and its detection limit is 0.2 µm, enabling it to selectively detect mercuric and cadmium ions in real samples using the MCR-ALS algorithm. In addition, the Opto sensor is non-toxic to the environment, biocompatible, reliable, low-cost, and produces reproducible results. Prabu and Mohamad [46] also designed a simple probe LC by incorporating curcumin with β-cyclodextrin for fluorescence sensing towards Hg^2+^ ions. The probe LC chemosensor was highly selective towards Hg^2+^ ions due to the highest change of absorption intensities by changing the solution color from yellow to colorless, which naked eyes can recognize. Besides, a strong “turn-off” fluorescence response with the highest percentage of quenching (73.88%) was observed after adding Hg^2+^ cation to probe LC. In contrast, no significant fluorescence quenching occurred for the other metal cations. A good linear correlation coefficient (R^2^ = 0.9971) was obtained using absorption spectroscopy in the sensitivity study of probe LC. When using the probe LC fluorescence chemosensor for the sensing of Hg^2+^ ions, the results showed a good linear correlation coefficient R^2^ = 0.9905 for Hg^2+^ ion concentration in the range from 0 to 45 µm, while the LOD and LOQ were 5.02 µm and 16.73 µm, respectively.

Other than that, Moussawi and Patra [47] used a simple aqueous phase method to produce nanostructured curcumin–zinc oxide hybrid materials for selective fluorescence sensing of As^3+^ ions and quick removal of As^3+^ ions from water without the oxidation and pH treatment. The nanostructured curcumin–zinc oxide found could significantly speed up the removal of arsenic contamination from the water below the maximum contaminant level within 30 min. Pourreza et al. [48] synthesized a paper-based analytical device by adding curcumin nanoparticles via wax-dipping technique to monitor the concentration of Hg^2+^ ions in various water sources. The authors explained that the absorption intensity of curcumin was directly decreased depending on the concentration of Hg^2+^ ions, causing the yellow color of the curcumin solution to gradually fade to a light-yellow color that naked eyes can observe. The experimental results showed that the linear range was 0.01–0.4 µg/mL of Hg^2+^ ions with the limit of detection of 0.003 µg/mL for the analyst.

Curcumin in food sensing applications offers some advantages such as simplicity, low cost, eco-friendliness, and the capability to yield colorimetric and fluorescence responses for specific or multiple metal ions depending on the experimental conditions. It also shows great potential to be used as sensing reagents in portable devices for rapid and real-time applications. However, the direct application of curcumin in sensors is hindered by some limitations, including its low water solubility, low bioavailability, short half-life, and rapid degradation under processing initiated by factors such as heat, light, and metallic ions, and oxygen [49]. Among the strategies developed to overcome the shortcomings of curcumin, nanoencapsulation has been addressed as one of the emerging and popular techniques [46,50,51,52]. Rafiee et al. [49] reviewed the nanoencapsulation strategies for loading curcumin. They classified them into lipid-based, chemical polymer- and biopolymer-based, nature-inspired, special equipment-based, and surfactant-based techniques.

### 2.8. Eggshells

Among various poultry wastes, eggshells have been of great interest in developing eggshell composites. The eggshells contain approximately 96% calcium carbonate and other organic materials [53]. Dayanidhi et al. [54] recently developed a colorimetric detector naturally derived from unmodified eggshell powder. The presence of calcium carbonate in the eggshell powder was revealed by the atomic fraction percentage of oxygen (53.5%), calcium (30.1%), and carbon (16.3%). The simple dispersion of eggshell powder in an aqueous solution of metal ions led to a color change from white to brown, pale green, pale blue, yellow, blue, pale pink, pale yellow, and dark yellow for different metal ions, which are Ag^+^, V^4+^, Cr^3+^, Cr^6+^, Cu^2+^, Co^2+^, Fe^2+^, and Fe^3+^ ions, respectively. Besides that, Mohammadi et al. [55] presented a novel method for differential-pulse anodic stripping voltammetric determination of Cd^2+^ ions using a carbon paste electrode modified with a nanosized magnetic nanocomposite of Fe_3_O_4_/eggshell and multi-walled carbon nanotubes. The magnetic nanoparticles exhibit specific properties such as less toxicity, good magnetic properties, large surface area-to-volume ratios, and ease of synthesis, coating, or functionalization, allowing them to be used as functional moieties for roughening the conductive sensing interface in electrochemical applications [56]. The response of the modified electrode to Cd^2+^ ions over the linear range from 3.0 to 250 ng/mL with a detection limit of 2.4 ng/mL for Cd^2+^ ions was obtained. Using waste material such as eggshells as a sensing material for heavy metals detection gives unique advantages, including its high availability, no cost, harmlessness to the environment, high thermal stability, and low density. The eggshells also show phase continuity in the composite and porous structure, which enables fewer materials to form a higher surface area than the artificial ones [55]. The particle aggregation can be minimized by using eggshells as bio-platforms to carry nanoparticles because of their porous structure with wide nucleation sites. However, the eggshell-based sensor is less selective than sensors that depend on other natural resources, such as sensing reagents, as it shows colorimetric responses to multiple metal ions.

### 2.9. Chlorophyll

A considerable interest has been given to the plant tissues and extracts in the development of colorimetric or fluorescence probes for detection, adsorption, and removal of residues in foods because these natural-derived sensors are all-natural, non-toxic, simple, and environmentally friendly as compared with the nanomaterials fabricated chemically in laboratories. Spinach is a widely consumed crop in the world and produces 25% of the waste from which essential biomolecules such as chlorophyll and phenolic compounds can be extracted [57]. Chlorophyll is the natural green pigment with amphiphilic nature present in most plants. Wang et al. [58] developed a new fluorescence sensor for detecting Hg^2+^ ions using peanut shells treated with alkali and modified with chlorophyll-a extracted from spinach leaves. In this study, both the fluorophore and the framework used to carry sensing materials were derived from natural plant tissues and extracts, contributing to the benefits of low cost due to the simplicity of the fabrication process, which is non-toxic and environmentally safe. The surface of alkali-treated peanut shells was modified with the chlorophyll-a extracted from leaves of spinach through simple adsorption. The photosynthesis pigment extracted from spinach leaves, chlorophyll-a, was used as a sensing reagent that Hg^2+^ ions can quench for fluorescence detection via absorption of pigment in the blue and, to a lesser extent, red portions of the light and emission of mainly red fluorescence. The chlorophyll-a was immobilized onto the peanut shell to induce red-shifted chlorophyll-a emission and enhance stokes shift, thereby improving the fluorescence quantum efficiency of chlorophyll-a and fluorescence quenching by Hg^2+^ ions. The cellulose and lignin in the peanut shells could act as excellent absorption reagents of heavy metal ions. The porous surface of the peanut shells becomes negatively charged via simple alkali treatment. Thus, it could further enhance the affinity of the peanut shell with cationic metal ions and increase the formation of interaction sites for the fluorophore and analysts on the surface of the peanut shell. The modified peanut shells could be used for sensitive fluorescence detection of Hg^2+^ ions over the linear range of 0–19 × 10^−8^ m with a detection lower limit of 8.5 × 10^−9^ m. The application of chlorophyll extracted from plants in the fluorescence sensor for heavy metal detection displays both pros and cons. Despite its high availability, low cost, ease of synthesis, and being harmless to humans and the environment, the limit of detection revealed by chlorophyll-based sensors is not low enough for practical application for trace detection of Hg^2+^ ions. By using free chlorophyll-a to detect Hg^2+^ ions in solution, there are other problems, including poor water solubility and pH stability of pigment, low quantum fluorescence, efficiency, photo-bleaching, and the spectral overlap between excitation and emission [58]. Immobilization of chlorophyll-a onto a kind of carrier material with a considerable surface area could be one of the alternatives to resolve these limitations. The natural nanotechnology-based sensor for detecting heavy metal ions is summarized and compiled in Table 1.

## 3. Biosynthesis of Natural Nanomaterials-Based

Synthesis of metallic nanoparticles has been appealing in nanotechnology and has been commonly applied in sensing applications in recent years. Due to simple, non-toxic, naturally available, and cost-effective benefits, enormous research on the green synthesis of metal nanoparticles using plant and waste biomaterials and their by-products are being investigated. The bio-organisms, especially plants and microorganisms such as algae, bacteria, fungi, or even viruses from the environment, are considered potential candidates for synthesizing nanomaterials due to their ability to accumulate and absorb metallic ions [10]. Moreover, the bioactive components, including amino acids, vitamins, phenols, and flavonoids present in biomaterials and biowaste materials, have physicochemical properties that play essential roles such as stabilizers, reducing, and capping agents in the synthesis of metallic nanoparticles [59].

### 3.1. Molasses

Molasses can be defined as dark and viscous runoff syrup produced from the crystallization stage of sugar cane and sugar beets refining industries [59]. Molasses is an eco-friendly and economically attractive energy source and consists of various bio-components such as glucose, vitamins, minerals, phenolic compounds, and non-sugar organic compounds. Moreover, molasses is composed of a high range of polyphenol content, possessing potent antioxidant capacity [60]. Manjari et al. [59] explored the usage of sugar cane secondary waste molasses as a biomaterial for synthesizing silver nanoparticles (AgNPs). The sugar molasses acted as a bio-component responsible for forming crystalline silver nanoparticles with oval shape, an average 16 nm size, and stability for 3 months. The formation of Ag^+^ to AgNPs using molasses as a reducing agent was observed at absorbance around 450 nm. The excitation of the surface plasmon resonance (SPR) band, which is connected to the morphology of the nanoparticles, caused the light brown color of the AgNPs solution to change to dark brown. The phenolic and amide groups could be involved in the synthesis and stabilization of AgNPs, while the carboxyl group and other compounds may bind to the surface of the AgNPs and act as a capping agent. The brown AgNPs solution became colorless after the addition of Hg^2+^ ions, whereas no significant color change was observed for the other metal ions such as Fe^2+^, Ca^2+^, Pb^2+^, Np^2+^, Cu^2+^, Mn^2+^, As^2+^, Co^2+^, Cd^2+^, Hg^2+^, Sn^2+^, and Cr^3+^ ions. The brown AgNPs solution was decolorized with a blue shift in the SPR band intensity due to the direct redox reaction between zero-valent Ag and Hg^2+^ ions in which the AgNPs were oxidized to Ag^+^ and Hg^2+^ ions were reduced to Hg atoms. The AgNPs synthesized were found to have a limit of detection at 0.025 µM concentrations of Hg^2+^ ions in real water bodies. They also exhibited good efficacy in detecting Hg^2+^ ions over the concentration range from 0.01 µM to 1 µM with a linear regression coefficient, R^2^ = 0.9809. The molasses-mediated synthesis of AgNPs offers a simple and cost-effective sensing method for monitoring metal ions in environmental fields for a sustainable future.

### 3.2. Green Plant, Fruit, and Herbs

Among these sustainable natural resources, the usage of the green plant-based leaf has acquired significance due to its availability, cost-effectiveness, and ability to function as both a reducing agent and capping agent to improve nanoparticle stability, thereby maximizing the efficiency of metallic nanoparticles extraction. Furthermore, the ease of synthesis makes the leaf extract preferable to be selected as a reducing agent for the preparation of metallic nanoparticles compared with other sources such as bacterial or fungal extract because of the requirement of aseptic conditions during the synthesis of nanoparticles.

For the first time, Roy et al. [61] reported the green synthesis of silver nanoparticles (AgNPs) from silver nitrate solution using leaf extract of *Dahlia pinnata*. They explored the rapid and selective colorimetric sensing activity of these biosynthesized nanoparticles. The functional biomolecules possibly reduced the silver ions and stabilized the particles in the mixture. The prepared nanoparticles were nearly spherical, with an average diameter of around 15 nm. More importantly, the biosynthesized AgNPs could selectively and instantly detect the presence of hazardous Hg^2+^ ions in water by changing color from light brown to colorless at the minimum detectable concentration of 10 µm within a wide range of pH. The mechanism of Hg^2+^ detection by suspension of AgNPs can be explained by the electrochemical differences between these metal ions: Hg^2+^ and Ag^+^ ions. The reduction potential of Ag^+^ is +0.80 V (Ag^+^ + e = Ag) whereas the standard reduction potential for Hg^2+^ is +0.92 V (2 Hg^2+^ + 2 e = Hg_2_^2+^) [62]. Hg^2+^ ions can oxidize the colloidal silver during interaction and form Ag^+^ ions, leading to the solution decolorization due to the higher reduction potential of the metals, signifying better oxidizing abilities according to the electrochemical series [63]. 

*Acalypha hispida* is known as a common flowering, evergreen, and annual shrub that can be found in tropical and subtropical regions. Based on the studies by Sithara et al. [64], *Acalypha hispida* leaf extract could be used for the synthesis of AgNPs with an average size of 20–50 nm and crystalline nature. The leaf extract added with methanol (MLE) was stirred with silver nitrate solution to produce a colorless mixture that changed its color to reddish-brown within 10 min, revealing the formation of AgNPs with surface plasmon resonance (SPR) peaks at 425 nm. Moreover, the natural components present in MLE acted as reducing agents to extract AgNPs and capping agents to improve the nanoparticle stability. When AgNPs solution interacted with Mn^2+^ ions, the color of the solution changed from reddish brown to colorless with suspended black particles. This is because of the receptor–ligand interaction between the capping agent produced from the MLE on AgNPs and the Mn^2+^ ions, leading to particle aggregation. No visible color change was observed for adding other metal ions, including Cr^3+^, Cr^6+^, Ni^2+^, Fe^2+^, Fe^3+^, and Zn^2+^ ions to the AgNPs solution. The minimum and maximum detection limits of developed AgNPs were 50 and 200 µm of Mn^2+^ ions, respectively.

Water hyacinths are invasive floating plants found in water bodies worldwide [65]. Previously, Oluwafemi et al. [66] synthesized AgNPs using hyacinth plant leaves for colorimetric sensing of Hg^2+^ in water solution. The results revealed that all the metals ions (Hg^2+^, Ba^2+^, Ca^2+^, Cr^3+^, Li^+^, K^+^, Co^2+^, Ni^2+^, Mn^2+^, Pb^2+^) reacted with AgNPs after adding them to the AgNPs solution. In contrast, the colorimetric sensing study indicated that the as-synthesized AgNPs responded well to heavy metal ions and selectively to Hg^2+^. Additionally, the Hg^2+^ ion was the only metal ion that caused a change in the color of the mixture from yellow to colorless when exposed to the AgNPs, indicating a selective response to Hg^2+^ ions compared to other metal ions. It creates a complex with the nanoparticle surface, which consists of negative charge ions from diverse functional groups linked to the AgNPs’ surfaces. These functional groups act as ligands; the electron from d-orbitals will repel when Hg^2+^ approaches them, causing the complex to change color. In addition, *Panax ginseng* is a popular herbal medicinal plant that is also known as Asian or Korean ginseng [67]. Several studies have reported using *Panax ginseng* root extract to synthesize metal nanoparticles. Tagad et al. [68] used *Panax ginseng* root extract to synthesize AgNPs and detect Hg^2+^. Research shows that *Panax ginseng* root extract can reduce and cap to form a stable colloidal suspension solution of AgNPs. The reactions of Ag with ginseng extract show a linear response of Hg^2+^ ranging from 10 µm to 1 mm with a limit of detection of 5 µm. Several other heavy metals were measured at 10 mm concentrations alongside Hg^2+^ to verify the sensor’s selectivity towards Hg^2+^. These metals include K^+^, Na^+^, Cu^2+^, Ni^2+^, Ca^2+^, Zn^2+^, Mg^2+^, and Mn^2+^. 

Fruit extracts have been found to contain high amounts of reducing agents [69]. There are many anthocyanins, ascorbic acid, flavonoids, and saccharides in fruits such as blueberries, blackberries, cornelian cherries, watermelon, grapes, arjuna, and pomegranates, which have been comprehended for synthesizing AgNPs [70]. A fruit-based synthesis of AgNPs has an additional advantage over synthesized AgNPs by biological methods. Recently, Firdaus et al. [71] synthesized AgNPs by utilizing an aqueous extract of *Carica papaya* fruit-assisted sunlight irradiation for colorimetric detection of Hg^2+^. After being turned into the yellowish-brown color of AgNPs, the solution turned colorless once Hg^2+^ was added. The detection was selective to Hg^2+^, and other heavy metals, including alkali and alkaline earth metals, did not change the color of AgNPs. The reduction–oxidation reaction results in the formation of AgNPs and the detection of Hg^2+^.

There has been great interest in synthesizing AgNPs using Chinese herbal extracts because of the potential synergistic effect between nanoparticles and capping agents [72]. Ji et al. [73] used *Radix Hedysari* (Chinese herb) to synthesize AgNPs that selectively detect Pb^2+^ in Yellow river samples. The AgNPs-RH exhibited great sensitivity for detecting Pb^2+^ with LOD of 1.5 µm, ranging from 10–500 µm. The mechanism is based on the interaction of hydroxyl radical with a carboxylate radical and Pb^2+^. Other than that, Chandraker et al. [74] utilized *Sonchus arvensis* leaf extract to synthesize AgNPs. The flavonoids, tannins, glycosides, and alkaloids were detected in the phytochemical analysis of *Sonchus arvensis* extract, indicating that these compounds may serve as capping, reducing, and stabilizing agents. The AgNPs obtained by this study were highly sensitive and selective to Fe^3+^ and Hg^2+^ with a limit of detection of 10^−3^ m. In the presence of sunlight, AgNPs-SA also showed strong catalytic efficiency against methylene blue, which completely degraded it within 1 h.

### 3.3. Frankincense Resin

Frankincense is an oleo gum resin derived from the Boswellia species, traditionally used to treat many diseases, mostly those dealing with memory functions [75]. Al Washahi et al. [76] recently established a green synthesis method of biogenic AuNPs from chloroauric acid (HAuCl_4_) using frankincense resin via microwave irradiation. The frankincense resin was composed of hydroxyl and carbonyl groups responsible for the synthesis and stabilization of AuNPs. The synthesized AuNPs colloidal solution showed red color, indicating the formation of AuNPs with a spherical shape with an average size distribution of 12 ± 2 nm and exhibited a strong absorption peak at 525 nm. The authors also developed a portable colorimetric probe of AuNPs that was selective towards Cu^2+^ ions with a color change from red to blue. The addition of Cu^2+^ ions to synthesized AuNPs solution led to a reduction in surface plasmon resonance (SPR) peak from 525 nm to 648 nm because of the complexation between the surface functional groups of AuNPs and Cu^2+^ ions. The developed colorimetric probe of AuNPs showed a good linear correlation, R^2^ = 0.92 over the concentration range of 50–550 nm of Cu^2+^ ions with a threshold detection of 50 nm at the optimum pH range from pH 4–8. The other metals such as Al^3+^, Mg^2+^, Ni^2+^, Sn^2+,^ Co^3+,^ Zn^2+^, Mn^2+^, and Hg^2+^ ions were not involved in the binding of Cu^2+^ ions to AuNPs. The developed AuNPs were suggested to be applied to detect Cu^2+^ ions in biomedical and environmental applications because they were cheap, simple, and portable.

### 3.4. Polysaccharides

Natural polymers, particularly polysaccharides, are extensively used to synthesize either homogeneous or heterogeneous nanostructures. As a result of their environmental safety, biodegradability, biocompatibility, renewability, intercomposition of multifunctional polymer chains, and ease of processing, different polysaccharides are ideal for dual action as stabilizers and reducing agents [77]. The polysaccharide obtained from Radix Hedysari has the benefits of high activity, less toxicity, no residue, and no drug resistance, which piqued researchers’ interest [78,79]. Previously, Chen et al. [80] reported the Hedysarum polysaccharides extracted from Radix Hedysari as reductants and green synthesis of AuNPs. Moreover, the large-sized AuNPs were used to selectively identify Fe^3+^ among the other metal ions in tap water with a range of 1 µm to 10 mM and a detection limit of 0.57 µm. The biogenic method of producing nanoparticles with well-defined size and morphology using plants is considered one of the best options for large-scale production. The gum karaya is a naturally available, cheap, and renewable highly branched polysaccharide that comes from plants of the *sterculia* species [81]. In the past, Gangapuram et al. [82] created a label-free colorimetric probe based on AuNPs for the sensitive and selective detection of Cu^2+^ in water samples. The AuNPs were synthesized using non-toxic polysaccharide carboxymethyl gum karaya, which functions as a reducing, stabilizing, and functionalizing agent. They are more stable to variables such as pH and salt and have a longer shelf life. These carboxymethyl gums, karaya-capped AuNPs, showed a selective colorimetric reaction to Cu^2+^ ions, with an observable color change from red to blue, which might be attributed to Cu^2+^ induced aggregation. The colorimetric probe system demonstrated an excellent linear correlation (R^2^ = 0.998) in the range of 10–1000 nm of Cu^2+^ with a detection limit of 10 nm under ideal conditions.

After cellulose, lignin is the second-most abundant naturally occurring biopolymer in the cell wall of lignocellulosic substances [83]. Lignin is a biomaterial with excellent properties, such as thermal stability and stiffness, and outstanding physical, chemical, and antioxidant properties [84]. Thus, lignin can replace hazardous chemicals in many chemical reactions as a reagent. Recently, Sajjadi et al. [85] showed that the layered structure of lignin contains numerous surface functional groups, such as carbonyl, hydroxyl, and phenolic, which reduce and stabilize noble metal ions. According to Yu et al. [86], using pulsed laser irradiation and the sonochemical process, a versatile green synthesis technique was used to synthesize functionalized AuNPs in lignin-based matrixes. Among the various metal ions, the L-AufNPs showed a highly selective colorimetric sensing ability toward Pb^2+^ ions. L-AufNPs display a prominent color change from red wine to purple when Pb^2+^ ions are detected, with a detection limit of 1.8 µm in the linear range of 0.1–1 mm. The L-AufNPs show greater selectivity towards Pb^2+^ than other metal ions due to the ionic strength of Pb^2+^ and the strength of the binding for Pb^2+^ to the carboxylic and phenolic types of sites [87,88].

### 3.5. Green Plant, Fruit Extract

*Impatiens balsamina,* also known as garden balsam, is an annual plant widely found in southern Asia. The leaf extract of *Impatiens balsamina* is traditionally applied in ailments of unwanted moles and other skin parts and treating snakebites. Furthermore, the leaf extract is rich in lawsone, lawsone methyl ether, and other organic compounds that possess a strong reducing ability, which has great potential to be utilized for the synthesis of metal nanoparticles [89]. Roy et al. [90] demonstrated the green synthesis of CuNPs by using leaf extract of *Impatiens balsamina*. After 24 h of incubation of leaf extract with copper sulfate solution, the solution showed light red color, indicating the formation of CuNPs because the leaf component, lawsone, acted as a reducing agent to convert Cu^2+^ ions to elemental metallic form in the medium. The color of the solution was then visually changed to dark red with a maximum absorption value of 550 nm observed after 72 h of incubation. The addition of Hg^2+^ ions caused the decolorization of dark red CuNPs solution at the detection limit of 1 ppm. Finally, the disappearance of the peak was observed at the concentration value of 10 ppm. For the addition of other metal ions, including Cr^3+^, Cd^2+^, Hg^2+^, Zn^2+^, Pb^2+^, and Fe^2+^ ions, there was no visual change observed, indicating the selectivity of synthesized copper nanoparticles toward Hg^2+^ ions. The mechanism of Hg^2+^ ions detection by synthesized CuNPs may be interpreted in light of the electrochemical differences between mercury and copper ions. Hg^2+^ ions with higher reduction potential and which tend to be reduced to metallic mercury would possibly oxidize the copper nanoparticles to Cu^2+^ during the interaction, leading to the decolorization of the suspension. Other metal ions could not oxidize copper nanoparticles due to their lower reduction potentials than Cu^2+^ ions [91]. In addition, these developed CuNPs can be an attractive alternative for wastewater management systems because of their ability to detect Hg^2+^ ions, low toxicity, cost-effectiveness, and strong photocatalytic property that allows the degradation of toxic organic dyes under solar irradiation.

In India, Aegle marmelos is one of the most prized medicinal trees [92], also called golden apple, emerald apple, and stone apple [93]. The peel extract from Aegle marmelos can be used as a reducing and capping agent to synthesize CuNPs. In a recent study, Kushwah et al. [94] described a noble green synthesis of CuNPs using Aegle marmelos fruit peel extract, describing their catalytic properties for sensitive chemiluminescence detection of Hg^2+^ ions in drinking water. Hg^2+^ was determined using changes in chemiluminescence in the presence of ethylenediamine. As a result, the ethylenediamine-CuNPs-Hg^2+^ complex promotes a higher rate of catalysis and consequently increases CL growth. Based on the given effect, Hg^2+^ concentration was determined by a calibration curve whose linear range is from 0.01 pm to 1.0 pm, with a detection limit of 0.0062 pm. Table 2 summarize the biosynthesis of metallic nanoparticles for identification of heavy metal ions.

## 4. Application of Engineered Nanomaterials

Nanomaterials have risen as an interesting new class of materials with various applications. The nanometer length could be illustrated by lining up five silicon atoms, each of which is one nanometer. Nanomaterials are defined as those whose size or one of their dimensions is between 1 and 100 nm [5]. Nanotechnology has developed novel materials with substantially unique features not found in bulk materials. Numerous studies have demonstrated nanomaterials’ huge surface area, surface-free energy, tiny size, active atomicity, and reactivity. Their large surface area-to-volume ratio enhances their solubility [95]. 

### 4.1. Quantum Dots

Throughout the years, material scientists have utilized various nanomaterials as sensitive sensors for identifying heavy metals in the water [96]. Nanostructures have generated considerable interest due to their ability to bridge the gap between the bulk and atomic levels. Due to their unique photochemical and photophysical capabilities, quantum dots (QDs) as zero-dimensional nanostructures have garnered considerable attention. According to the quantum mechanics’ wave-particle duality hypothesis, QDs are nanoscale crystal materials with characteristics between bulk materials and discrete atoms [97]. According to Chen et al. [98], a simple and environmentally acceptable technique for producing water-dispersible N-doped SiQDs using wheat straw and AMIMCl has been developed. Due to their high toxicity, silane-based precursors for producing water-dispersible silicon quantum dots (SiQDs) pose a risk to researchers and the environment. Due to their excellent photoluminescence stability, acceptable biocompatibility, and low toxicity, silicon quantum dots (SiQDs) have been widely investigated for bioimaging and biosensing [99]. Wheat silicon quantum dots (WS–SiQDs) exhibited homogeneous spherical morphologies, N-doped structures, and strong fluorescence emission with a quantum yield of 28.9%. WS–SiQDs@silica hydrogels showed increased sensitivity for detecting Cr(VI) and Fe^3+^ in water, with detection limits as low as 142 and 175 nm, respectively. Gan et al. [100] successfully developed a strategy for ultrasensitive Hg^2+^ detection using carbon quantum dots modified with europium complexes (CQDs@Ad–Eu–DPA). CQDa@Ad–Eu–DPA exhibited dual-emission wavelengths by incorporating blue emissive carbon quantum dots (CQDs) with red emissive europium complexes (Ad–Eu–DPA). The adenine (Ad), dipicolinic acid (DPA), and europium ion (Eu^3+^) were used to create the Ad–Eu–DPA. CQDs were synthesized using citric acid and triethylenetetramine as source materials. To detect Hg^2+^, the fluorescence enhancement of the CQDs@Ad–Eu–DPA technique was designed. CQDs@Ad–Eu–DPA emitted light at two distinct wavelengths of 443 and 617 nm. The fluorescence sensor displayed outstanding sensitivity and selectivity for Hg^2+^ detection, with a detection limit of 0.2 nm and a detection range of 1–20 nm. Additionally, the new approach identified Hg^2+^ in drinking water and milk samples with satisfactory recovery rates (97.6–105.4%). This technique is well suited for ultrasensitive detection of Hg^2+^ at extremely low concentrations.

Preparing highly luminous nitrogen-doped carbon quantum dots (N–CQDs) using chitosan as both the carbon and nitrogen source was simple, affordable, and a one-step hydrothermal technique. The as-prepared N–CQDs have an average size of 2 nm and display wavelength-dependent fluorescence, with maximal excitation and emission wavelengths of 330 and 410 nm, respectively. They were used to determine Fe3+ concentrations in tap water and lake water samples to assess the practical capability of N–CQD colorimetric probes. Based on the remarkable fluorescence quenching capability of Fe^3+^, the N–CQDs demonstrated exceptional selectivity and sensitivity. They were effectively employed for the quantitative detection of Fe^3+^ in aqueous solutions with a linear detection range of 0–500 m and detection limit of 0.15 m. This approach is simple, quick, and efficient. It may be used to detect Fe^3+^ [101].

Fu et al. [102] invented a new technique of electrochemical detection of silver ions in water samples utilizing sulfur quantum dots (SQDs) modified gold electrodes. Because of the strong affinity between Au and S atoms, the SQDs were synthesized top-down by alkali etching of bulk powder and then immediately employed for Au electrode surface modification. The produced SQDs had an average size of 3.85 nm and were well dispersible, allowing them to be directly employed for Au electrode surface modification. Due to the extreme affinity between Au and S atoms, the SQDs were effectively deposited on the electrode surface. Despite SQDs reducing the electrode’s electroconductivity, the high affinity between S and Ag makes it particularly appropriate for Ag^+^ sensing due to the tiny size of SQDs. The suggested sensor has a linear detection range of 0.1 nm^−3^ m under ideal conditions. The detection limit of Ag^+^ was 71 pm, and this approach is viable for detection.

### 4.2. Carbon Nanotubes

Carbon nanotubes (CNT) are infinite cylindrical graphitic sheets with sp^2^ hybridization that have been organized into a signal network with hexagonal cells. The presence of a polar functional group on the surface of a nanotube plays an important role in improving adsorbent performance. Heavy metal removal effectiveness is highly dependent on surface total acidity. It has no positive link with CNT type, a specific pore volume, mean pore diameter, or specific surface area. The adsorption process in CNT is typically regulated by the hollow interior of distinct CNTs labeled as internal sites, the interstitial channels between separate CNTs in stacks, the grooves present on the side of a CNT stack, and the exterior surface, and the external surface of individual CNTs [103].

Palisoc et al. [104] used discarded batteries to create a novel approach for detecting heavy metals in herbal dietary supplement samples using bismuth/multi-walled carbon nanotubes/nafion modified graphite electrodes. The crop-coating approach was used to modify the graphite electrode using bismuth nanoparticles (BiNP), multi-walled carbon nanotubes (MWCNT), and Nafion. Anodic stripping voltammetry was used to determine trace levels of Cd^2+^ and Pb^2+^ using bare and altered graphite electrodes as the working electrode. The modified electrode demonstrated superior electroanalytical performance for detecting heavy metals compared with the bare graphite electrode. At optimized experimental settings and parameters, a linear concentration range of 5 parts per billion (ppb) to 1000 ppb (R^2^ = 0.996) and LOD of 1.06 ppb for Cd^2+^ and 0.72 ppb for Pb^2+^ were achieved.

Feist and Sitko [105] conducted another experiment in which they detected Pb, Cd, Zn, Mn, and Fe in rice samples using carbon nanotubes and cationic complexes of bathophenanthroline and oxidized multiwalled carbon nanotubes (ox-MWCNTs) were studied as a new sorbent for the simultaneous preconcentration of trace levels of Pb^2+^, Cd^2+^, Zn^2+^, Mn^2+^, and Fe^3+^ with bathophenanthroline using a chelating agent impregnated on their surface. The approach that has been developed is based on dispersive micro-solid phase extraction (DMSPE). This approach enables quick contact between analytes and adsorbent particles, significantly reducing sample preparation time compared with conventional SPE. Cd, Pb, Zn, Mn, and Fe have detection limits of 0.13, 0.25, 0.24, 0.32, and 0.35 ng/mL, respectively. These heavy metals may be identified in the presence of alkaline metals. Lu et al. [106] demonstrated a controllable electrochemical response of electro synthesized layer-by-layer multilayer films based on multi-walled carbon nanotubes and metal-organic frameworks as a high-performance electrochemical sensor for simultaneous cadmium and lead measurement. The electrode’s surface was modified using various multilayer films created by electrodeposited carboxylated multi-walled carbon nanotubes (MWCNTs-COOH) and Zr-metal-organic frameworks (UiO-66-NH_2_). Calculated detection limits for Cd^2+^ and Pb^2+^ were 0.09 ppb and 0.071 ppb. A simultaneous electrochemical sensor was developed using Fe_3_O_4_ nanoparticles, Fe_3_O4/multi-walled carbon nanotubes (Fe_3_O_4_/MWCNTs), and Fe_3_O_4_/fluorinated multi-walled carbon nanotubes (Fe_3_O_4_/F-MWCNTs) nanocomposites, all of which were synthesized by hydrothermal method and tested in real river water and soybean samples using square wave anodic stripping voltammetry (SWASV). The sensitivity of Fe_3_O_4_/F-MWCNTs was higher than that of Fe_3_O_4_/MWCNTs or Fe_3_O_4_, and the LODs of the Fe_3_O_4_/F-MWCNTs sensor were 0.05, 0.08, 0.02, and 0.05 nm for Cd^2+^, Pb^2+^, Cu^2+^, and Hg^2+^, respectively. Due to its exceptional performance in selectivity, stability, and repeatability, this Fe_3_O_4_/MWCNTs sensor can be used in various applications [107].

### 4.3. Nanogels

Nanogels are nanoparticles formed of a hydrogel that has been extensively cross-linked, either physically or chemically, with hydrophilic polymer chains. Due to the obvious presence of hydrophilic functional groups in nanogels, they can store a significant quantity of water. They can swell in excellent solvents while retaining their internal structures and characteristics [108]. In order to visually detect traces of lead dioxide (Pb^2+^), a simple and portable thermometer-type instrument focused on forward osmosis-driven liquid column rising is being created [109]. The device comprises a top indicator tube, a chamber containing Pb^2+^-responsive poly (N-isopropylacrylamide-co-benzo-18-crown-6-acrylamide) smart nanogels (PNB), and a bottom semipermeable membrane. Pb^2+^ has a detection limit of 10^−10^ m. Compared with previous methods for detecting Pb^2+^, this study offers a simple and portable device that circumvents the problems associated with the preparation and operation of existing technologies [110,111,112,113]. Wang et al. [114] developed a unique smart membrane with ion-recognizable nanogels acting as gates in linked pores to detect trace Pb^2+^ ions in water. The membrane was prepared in a single step using the vapor-induced phase separation (VIPS) method. It features an interconnected porous structure with gates made of Pb^2+^-responsive poly(N-isopropylacrylamide-co-benzo-18-crown-6-acrylamide) (PNB) nanogels on polyethersulfone (PES) membrane pore surfaces. Due to the great sensitivity and selectivity of PNB nanogels on the membrane pore surfaces in reaction to Pb^2+^ ions, the produced membranes can detect Pb^2+^ ions efficiently with the LOD of 10^−9^ mol/L.

### 4.4. Dendrimer

Dendrimers are monodisperse, highly branched, three-dimensional “ball-like” polymers that can grow in size step-by-step through repeated reactions. They are a group of macromolecules with many surface groups, a small shape, and empty spaces inside that can be used to hold gust molecules. In the mid-1980s, Tomalia et al. [115] synthesized the first dendrimers. Since then, several types of dendrimers have been synthesized, such as fluorescent dendrimers, which are synthesized by covalently attaching fluorophores to the dendrimer structure. Dendrimers have several unique characteristics, including the capacity to trap tiny molecules, which makes them suitable for biological applications, including nanoreactors and catalysis [116,117,118,119,120,121]. Adam et al. [122] sought to determine the capacity of the modified PAMAM dendrimers to interact with metal ions Zn^2+^, Cd^2+^, and Hg^2+^. The researchers synthesized and used a PAMAM dendrimer with 4-N,N’-dimethylethylenediamine-1,8-naphthalmide units as chromophores at its rim to absorb dangerous heavy metal ions. This modified fluorescent dendrimer (FCD) was complexed in a 2:1 ratio with Group 12 metal ions. Fluorescence and absorption spectroscopy was used to analyze the photophysical characteristics of the FCD molecule and its metal complexes. The absorption and emission spectra of the FCD molecule following complexation with the Zn^2+^ ion exhibit larger spectrum alterations than those of the other two metal ions. Another by Maleki et al. [123] synthesized and evaluated second-generation polyamidoamine dendrimer-functionalized magnetic nanoparticles (Fe_3_O_4_@G2-PAD) to detect Pb^2+^ and Cd^2+^ ions in ambient fluids. Simultaneous determination of analyte cations was performed using square wave anodic stripping voltammetry (SWASV). The researchers examined and optimized the effect of experimental factors on the performance of the modified magnetic electrode. Due to the Fe_3_O_4_@G2-PAD nanocomposite’s high surface-to-volume ratio and high adsorption capacity, the improved MCPE exhibited good electrochemical characteristics for the simultaneous detection of Pb^2+^ and Cd^2+^ via SWASV. Under ideal conditions, the value of the limit of detection (0.17 mg mL^−1^ for Pb and 0.21 mg mL^−1^ for Cd) was under the WHO-recommended threshold limits (10 ng mL^−1^ for Pb and 3 ng mL^−1^ for Cd) for cadmium and lead in drinking water and this sensor was successfully used to measure Pb^2+^ and Cd^2+^ ions in a variety of real water samples.

A sensitive and efficient method for the simultaneous rapid determination of cadmium and mercury ions in water samples was developed using magnetic PAMAM dendrimers as sorbents for magnetic solid-phase extraction (MSPE) in conjunction with high-performance liquid-phase chromatography and an ultraviolet variable wavelength detector (HPLC-VWD). During the elution procedure, sodium diethyldithiocarbamate (DDTC-Na) was utilized as a chelating agent, and Cd^2+^ and Hg^2+^ LODs were as low as 0.016 g L^−1^ and 0.040 g L^−1^, respectively. This approach showed several advantages, including low cost, rapidity, adequate sensitivity, and reusability [124]. Baghayeri et al. [125] proposed the synthesis, classification, and application of an electrochemical sensor for the detection of As^3+^ at trace levels in water samples in which magnetic multiwall carbon nanotubes covalently functionalized with PAMAM dendrimer (MMWCNTs-D-NH_2_) were produced, generally characterized and presented as recognition elements in sensor preparation. The electrochemical properties of the invented sensor were investigated using square wave anodic stripping voltammetry (SWASV) measurements at various As ^3+^ concentrations. The sensor demonstrated excellent sensitivity and selectivity against various probable interferents, including Hg^2+^, Pb^2+^, Cd^2+^, and Cu^2+^. The limit of detection of As^3+^ was determined to be 0.46 µg L^−1^.

### 4.5. Iron Oxide Nanoparticles

Several metal oxides have been investigated, but iron oxide (Fe_3_O_4_) NPs have received the most attention because of their strong affinity for heavy metal ions, chemical stability, nontoxicity, and cost efficiency. Due to these features, Fe_3_O_4_-based NPs, particularly those composited with reduced graphene oxide (RGO) and room temperature ionic liquid, γ-AlOOH (boehmite), have been successfully employed as electrochemical sensors [126,127,128]. According to the results of a study by Deshmukh et al. [129], an iron oxide (Fe_3_O_4_) nanoparticle (NP) with terephthalic acid (TA) cap was synthesized by simple wet chemical synthesis. The NPs were then used to detect ions of Hg, Pb, and Cd in water, individually and simultaneously, and selectively using an electrochemical approach. Under optimized experimental circumstances, the limit of detection of Hg^2+^, Pb^2+^, and Cd^2+^ ions was 0.1 µm, 0.05 µm, and 0.01 µm, respectively, for individual analysis, whereas the LOD values for Hg^2+^, Pb^2+^, and Cd^2+^ ions were 0.3 µm, 0.04 µm, and 0.2 µm for simultaneous analysis. Lee et al. [130] provided an analytical evaluation of an iron oxide (Fe_2_O_3_)/graphene (G) nanocomposite electrode utilized in association with in situ plated bismuth (Bi) to determine Zn^2+^, Cd^2+^, and Pb^2+^. The electrochemical characteristics of a modified Fe_2_O_3_/G/Bi composite electrode were examined using differential pulse anodic stripping voltammetry to identify heavy metal ions. Due to the synergetic impact of graphene and Fe_2_O_3_ nanoparticles, the modified electrode demonstrated enhanced electrochemical catalytic activity and sensitivity to trace heavy metal ions. Under optimized circumstances, the electrode’s linear range was 1–100 g L^−1^ for Zn^2+^, Cd^2+^, and Pb^2+^, with LODs of 0.11 g L^−1^, 0.08 g L^−1^, and 0.07 g L^−1^, respectively. The solventless thermal decomposition approach used to create simple and easy nanocomposite electrode materials may be expanded to create nanocomposites and promising electrode materials for heavy metal ion detection.

### 4.6. Gold Nanoparticles

Gold is a chemical element with the symbol Au, derived from the Latin word aurum with an atomic number 79, and is the transition element of Group 11 elements [2]. Gold nanoparticles (AuNPs) have shown a lot of potential for building biosensors that can help identify food safety [131]. He et al. [132] created a sensitive biosensor for identifying mercury ion based on hairpin hindrance by thymine–Hg^2+^–thymine structure, where the double-amplified electrochemical DNA Hg^2+^ detector was created. The sensor comprises two major components: a DNA-functionalized Au nanoparticle and an electrode modified with a nanocomposite. The DNA-Functionalized Au nanoparticle consisted of hairpin probe DNA and linear signal DNA that functioned as a signal trigger. With a LOD of 0.21 pm, the avidin/AuNps/NGP/Nafion/GCE system worked well in analyzing Hg ions in Chinese herbs. By varying the sequences of the enzyme/substrate strands, the biosensor may also detect divalent metal ions such as Pd^2+^ and Cu^2+^.

Sadani et al. [133] developed LSPR-based U-bend optical fiber sensor for mercury identification. The sensor is inspired by the very causes of mercury bioaccumulation in life forms, namely its proclivity to form complexes with proteins and its ability to behave as a weak acid with a strong affinity for sulfur and phosphorus-containing moieties, which are abundantly present in proteins and polysaccharides of animal origin. The crucial innovation is the combination of bovine serum albumin (a protein) and chitosan-capped gold nanoparticles for mercuric ion detection; additionally, the chemistries used to integrate the receptors result in a shallow noise floor sensor. The LOD was determined to be 0.1 ppb in tap water. The Au nanoprobe was created via functionalization due to its biological significance and chelating properties. The researchers created a new 2-thiazoline-2-thiol functionalized gold (Au-TT) nanosensor for the selective and sensitive detection of hazardous heavy metal ions, Hg^2+^ and Pb^2+^, using colorimetry, as well as for the evaluation of its competitive surface reactivity via SERS and XPS. The detection method and surface reactivity are based on the thiocarbonyl S or thiazoline ring N/S atom’s competitive binding affinity for metal ions and nanoparticles. Hg^2+^ and Pb^2+^ had a LOD of 0.111 ppm and 0.096 ppm, respectively. Additionally, the Au-TT nanosensor’s selectivity and specificity were evaluated in samples from various water sources and the influence of interference from various metal ions. The Au-TT biosensor was shown to be selective for Hg^2+^ and Pb^2+^ in actual samples and high specificity for Pb^2+^ in the presence of other metal ions [134]. 

By cyclic voltammetry deposition, a Bi/Nafion/RGO-GNPs/GCE composite-modified GCE was produced and subsequently explored for the simultaneous determination of Cd^2+^ and Pb^2+^ using SWASV [135]. SEM, CV, and SWASV were used to characterize the manufactured Nafion/RGO-GNPs/GCE. The parameters such as RGO-GNP electrodeposition cycles, pH value, Bi^3+^ concentration, deposition potential, and deposition duration were optimized. Lower cycles of RGO-GNPs deposition were favorable for avoiding GNP aggregation. The RGO in the Nafion/RGO-GNPs/GCE improved the electrodes’ total surface area, electrical conductivity, and dispersion of the GNPs. At the same time, the GNPs enhanced the active sites for redox reactions and increased the electrodes’ electron transfer kinetics. Furthermore, by combining the co-deposit ability of bismuth film with heavy metals and the ion-exchange property of Nafion film, the produced electrode demonstrated extremely high sensitivity in the measurement of Cd^2+^ and Pb^2+^. In the simultaneous detection of Cd^2+^ and Pb^2+^ in the concentration range of 1.0 to 90 gL^−1^, the Bi/Nafion/RGO-GNPs/GCE displayed a highly linear behavior LOD of 0.08 g L^−1^ and 0.12 g L^−1^, respectively. Table 3 shown the types of engineered nanomaterials, such as quantum dots, carbon nanotubes, nanogels, dendrimer, iron oxide, and gold nanoparticles, used to identify heavy metal ions.

## 5. Conclusions and Future Perspectives

With the advancement of technologies in environmental applications, nanotechnology has been the focus of researchers to improve environmental safety, especially the development of sensors for the detection of various contaminants present in soil, water, and agricultural products. This overview of the plant-mediated green carbon dots and their recent progress in the optical detection of major heavy metal ions revealed their escalated development in recent years. However, some challenges still need to be addressed for their possible scalability and application as economically viable daily life sensing probes. Apart from the synthesis strategies for developing highly stable and efficient green nanomaterials, the emission from the entire visible spectrum and narrow bandwidth of fluorescence signal is required for specific applications and enhanced sensitivity. Thus, we believe that the upcoming exploration of sensitive optical detection systems using carbon dots will gain extensive attention in food, agriculture, and textile pollutant sensing because of its simplicity, biocompatibility, and cost-effectiveness.

## Figures and Tables

**Figure 1 nanomaterials-12-02665-f001:**
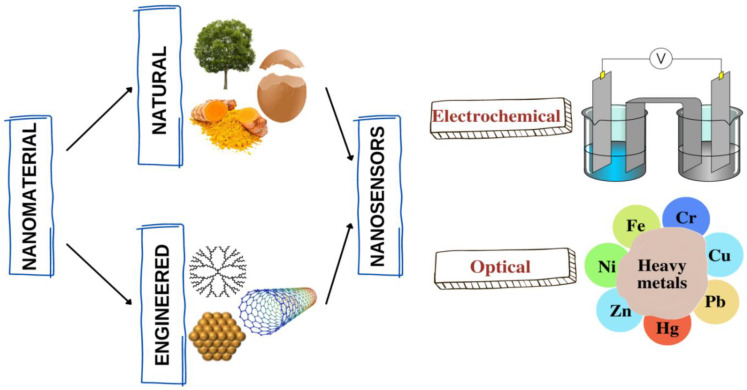
Natural and engineered nanomaterials for sensing metal ions.

**Table 1 nanomaterials-12-02665-t001:** Natural nanotechnology-based sensor for detection of heavy metal ions.

Nanomaterials	Detection Technique	Analyst	Sample	Limit of Detection (LOD)	Reference
Lotus root CDs	Fluorescence	Hg^2+^	Tap water	18.7 nm	[16]
Rose-heart radish CDs	Fluorescence	Fe^3+^	River water	0.13 µm	[17]
Red lentils CDs	Fluorescence	Fe^3+^	Water	0.10 µm	[18]
Coconut coir CDs	Fluorescence (turn-on)	Cd^2+^	Deionized water, tap water, sewage water, and groundwater	0.18 nm	[20]
	Fluorescence (turn-off)	Cu^2+^		0.28 nm	
Prawn shells CDs	Fluorescence	Cu^2+^	Drinking water, river water and sea water	5 nm	[15]
Gardenia fruit CDs	Fluorescence (turn-off)	Hg^2+^	Environmental and human samples	320 nm	[24]
*Poa pratensis* L. CDs	Fluorescence	Fe^3+^, Mn^2+^	Water	1.4 µm, 1.2 µm	[29]
Wintersweet flower CDs	Fluorescence (Dual mode)	Cr(VI), Fe^3+^	Water	0.07 µm, 0.15 µm	[30]
Carbon nano-onions from flaxseed oil	Fluorescence	Al^3+^	Wastewater	0.77 µm	[35]
Curcumin	Colorimetric	Pb^2+^	Rice	0.9 µm	[41]
Curcumin and Aloe vera	Colorimetric	Fe^3+^	Water	27.84 ppm	[42]
Curcumin	Colorimetric	Pb^2+^	Water	1 mm	[43]
Curcumin	Colorimetric	Pb^2+^	Water	20 µm	[44]
Curcumin, anthocyanin	Colorimetric	Hg^2+^, Cd^2+^	River water	0.2 µm	[45]
Curcumin	Fluorescence (turn-off)	Hg^2+^	Drinking water and tap water	5.02 µm	[46]
Curcumin	Fluorescence	As^3+^	Water	100 ppb	[47]
Curcumin	Colorimetric	Hg^2+^	Environmental and industrial water	0.17 µg/mL	[48]
Eggshell	Colorimetric, adsorption	Fe^2+^, Fe^3+^, Cu^2+^ Ag^+^, Cr^3+^, Cr^6+^, Cu^2+^, Co^2+^ V^4+^	Wastewater	10^−4^ m 10^−3^ m 5 × 10^−3^ m	[54]
Eggshell/Fe_3_O_4_	Electrochemical, voltammetric	Cd^2+^	Water	2.4 ng mL^−1^	[55]
Chlorophyll (spinach)	Fluorescence	Hg^2+^	Lake water	8.5 × 10^−9^ m	[58]

**Table 2 nanomaterials-12-02665-t002:** Biosynthesis of metallic nanoparticles for identification of heavy metal ions.

Biosynthesized Metallic Nanoparticles	Natural Resources	Analyst	Detection Technique	Samples	Limit of Detection (LOD)	Reference
AgNPs	Molasses	Hg^2+^	Colorimetric	River water, drinking water and tap water	0.025 µm	[59]
	*Dahlia pinnata* leaf extract	Hg^2+^	Colorimetric	Water	10 µm	[61]
	*Acalypha hispida* leaf extract	Mn^2+^	Colorimetric	Wastewater (Iron and steel industry effluent)	50 µm	[64]
	Water hyacinth plant leaves	Hg^2+^	Colorimetric	Water	-	[66]
	*Panax ginseng* root	Hg^2+^	Colorimetric	Water	5 µm	[68]
	*Carica papaya* fruit	Hg^2+^	Colorimetric	Water	-	[71]
	*Radix Hedysari*	Pb^2+^	Colorimetric	Yellow river water (China)	1.5 µm	[73]
	*Sonchus arvensis* leaf extract	Fe^3+^, Hg^2+^	Colorimetric	Water	10^−3^ m	[74]
AuNPs	*Frankincense* resin	Cu^2+^	Colorimetric	Bore water and tap water	50 nm	[76]
	*Hedysarum* polysaccharides	Fe^3+^	Colorimetric	Tap water	0.57 µm	[80]
	Carboxymethyl gum karaya	Cu^2+^	Colorimetric	Tap water, river water and bore water	10 nm	[82]
	Lignin	Pd^2+^	Colorimetric	Water	1.8 µm	[86]
CuNPs	*Impatiens balsamina* leaf extract	Hg^2+^	Colorimetric	Water	1 ppm	[90]
	*Aegle marmelos* fruit peel extract	Hg^2+^	Chemiluminescence	Drinking Water	0.0062 pm	[94]

**Table 3 nanomaterials-12-02665-t003:** Engineered nanomaterials for identification of heavy metals.

Types of Nanomaterials	Nanosensor	Analyst	Detection Technique	Samples	Limit of Detection (LOD)	Reference
Quantum Dots	WS-SiQDs@silica hydrogels	Cr(VI), Fe^3^	Fluorescence	Water	142 nm, 175 nm	[98]
	CQDs@Ad-Eu-DPA	Hg^2+^	Fluorescence	Drinking water	0.2 nm	[100]
	N-CQDs	Fe^3+^	Fluorescence	Tap water	0.15 m	[101]
	SQDs	Ag	Electrochemical	Water	71 pm	[102]
Carbon nanotubes	BiNP/MWCNT/Nafion modified graphite electrode	Cd^2+^, Pb^2+^	Electrochemical	Herbal dietary supplement	1.06 ppb, 0.72 pp	[104]
	ox-MWCNTs	Cd^2+^, Pb^2+^, Zn^2+^, Mn^2+^, Fe^2+^	Electrochemical	Rice	0.13 ng/mL, 0.25 ng/mL, 0.24 ng/mL, 0.32 ng/mL, 0.35 ng/mL	[105]
	MWCNTs/UiO-66-NH_2_/MWCNTs/COOH	Cd^2+^, Pb^2+^	Electrochemical	Water	0.09 ppb, 0.071 ppb	[106]
	Fe_3_O_4_/F-MWCNTs	Cd^2+^, Pb^2+^, Cu^2+^, Hg^2+^	Electrochemical	Soybean	0.05 nm, 0.08 nm, 0.02 nm, 0.05 nm	[107]
Nanogels	PNB Nanogels	Pb^2+^	Colorimetric	-	10^−10^ m	[109]
	Smart membrane PNB nanogels	Pb^2+^	Colorimetric	Water	10^−9^ mol L^−1^	[114]
Dendrimers	Modified fluorescence dendrimers (FCD)	Zn^2+^, Cd^2+^, Hg^2+^	Fluorescence	-	-	[122]
	Fe_3_O_4_@G2-PAD	Pb^2+^, Cd^2+^	Electrochemical	Water	0.17 ng mL^−1^, 0.21 ng mL^−1^	[123]
	PAMAM dendrimers	Cd^2+^, Hg^2+^	Electrochemical	Water	0.016 g L^−1^, 0.040 g L^−1^	[124]
	MMWCNTs-D-NH_2_	As^3+^	Electrochemical	Water	0.46 µg L^−1^	[125]
Iron Oxide Nanoparticles	TA/Fe_3_O_4_	Hg^2+^, Pb^2+^, Cd^2+^	Electrochemical	Water	0.3 µm, 0.04 µm, 0.2 µm	[129]
	Fe_2_O_3_/G/Bi	Zn^2+^, Cd^2+^, Pb^2+^	Electrochemical	-	0.11 g L^−1^ 0.08 g L^−1^, 0.07 g L^−1^	[130]
Gold nanoparticles	avidin/AuNps/NGP/Nafion/GCE	Hg ^2+^	Electrochemical	Chinese herbs	0.21 pm	[132]
	LSPR based U-bend optical fiber	Hg^2+^	Colorimetric	Water	0.1 ppb	[133]
	2-thiazoline-2-thiol functionalized gold (Au-TT)	Hg^2+^, Pb^2+^	Colorimetric	Water	0.111 ppm, 0.096 ppm	[134]
	Bi/Nafion/RGO-GNPs/GCE	Cd^2+^, Pb^2+^	Electrochemical	Soil	0.08 g L^−1^, 0.12 g L^−1^	[135]

## Data Availability

Not applicable.

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
