# Peer review of "Natural and Engineered Nanomaterials for the Identification of Heavy Metal Ions—A Review"

_nanomaterials, 2022, doi:10.3390/nano12152665_

Round 1

Reviewer 1 Report

Paper entitled “Natural and Engineered Nanomaterials for Identification of Heavy Metals - A Reviewmeets the necessary standards for publication in this journal.

Attention when writing references. They are not unitary.

Final Conclusion: The paper meets the necessary standards for publication.

Author Response

Reviewer 1:

The paper entitled “Natural and Engineered Nanomaterials for Identification of Heavy Metals - A Review” meets the necessary standards for publication in this journal.

Attention when writing references. They are not unitary.

Final Conclusion: The paper meets the necessary standards for publication.

  • Thank you very much for your comments and suggestions.

Reviewer 2 Report

This review emphasizes the naturally derived and engineered nanomaterials that have the potential to be applied as sensing reagents to interact with metal ions or reducing and stabilizing agents to synthesize metallic nanoparticles for the detection of heavy metals. The synthesis method and detection techniques of each potential natural resource for heavy metal detection are also highlighted and concluded with a brief discussion regarding their pros and cons for sensing applications. This topic is interesting and attractive to the readers. The manuscript is acceptable after major revision by solving the following issues. 

1.     To be specifically, the statements like “detection of heavy metals” should be revised as “detection of heavy metal ions”.

2.     It would be better to replace the keywords of “contaminants” by “heavy metal ions”.

3.     The authors should pay attention to the spelling of some words, for example “emphasises”, “synthesise”, “stabilising”, etc. need to be revised.

4.     To be strict, fluorescence is a kind of optical phenomenon/technology. Hence, the Figure 1 needs to be revised.

5.     Biomass derived carbon materials show promising applications in the water treatment including removing heavy metal ions. Some references are recommended, for example Journal of Bioresources and Bioproducts 2020, 5, 238-247; Journal of Bioresources and Bioproducts 2020, 5, 204-210.

6.     The contents of “3. Green synthesis of metallic nanoparticles using natural resources” and “4. Engineered nanomaterials in detection of heavy metals” are partly overlay. Please rewrite part of them.

7.     The authors should pay attention to the unsuitable statements like “The present work synthesised and evaluated second-generation polyamidoamine dendrimer-functionalized magnetic nanoparticles (Fe3O4@G2-PAD) for the detection of Pb2+ and Cd2+ ions in ambient fluids.” It is confusing to say “The present work…”. Some review papers are recommended for references: Coordination Chemistry Reviews 2022, 466, 214604; Journal of Bioresources and Bioproducts 2021, 6, 292-322; Nanoscale 2022, 14, 8216.

8.     Some examples are not related to the topic of “natural”, please remove them.

9.     The conclusion part is too short. Some insightful conclusions and prospects should be offered.

10.  The authors should pay attention to the formation of references, especially the writing of superscripts and subscripts.

Author Response

Reviewer 2:

This review emphasizes the naturally derived and engineered nanomaterials that have the potential to be applied as sensing reagents to interact with metal ions or reducing and stabilizing agents to synthesize metallic nanoparticles for the detection of heavy metals. The synthesis method and detection techniques of each potential natural resource for heavy metal detection are also highlighted and concluded with a brief discussion regarding their pros and cons for sensing applications. This topic is interesting and attractive to the readers. The manuscript is acceptable after major revision by solving the following issues. 

  • Thank you very much for your comments and suggestions.
  1.     To be specifically, the statements like “detection of heavy metals” should be revised as “detection of heavy metal ions”.
  •  The authors has revised the statements in the entire manuscript.
  1.     It would be better to replace the keywords of “contaminants” by “heavy metal ions”.
  • The keywords have been replaced.
  1.     The authors should pay attention to the spelling of some words, for example “emphasises”, “synthesise”, “stabilising”, etc. need to be revised.
  • The authors have been revised the spelling of some words in the manuscript.
  1.     To be strict, fluorescence is a kind of optical phenomenon/technology. Hence, the Figure 1 needs to be revised.
  • The figure has been revised accordingly.
  1.     Biomass derived carbon materials show promising applications in the water treatment including removing heavy metal ions. Some references are recommended, for example Journal of Bioresources and Bioproducts 2020, 5, 238-247; Journal of Bioresources and Bioproducts 2020, 5, 204-210.
  • Thank you very much for the suggestions. However, the authors only cited related references in the manuscript.
  1.     The contents of “3. Green synthesis of metallic nanoparticles using natural resources” and “4. Engineered nanomaterials in detection of heavy metals” are partly overlay. Please rewrite part of them.
  • The authors has revised the subtopics for Part 3. Part 3 focused on the natural resources used for green synthesis of metallic nanoparticles. 
  1.     The authors should pay attention to the unsuitable statements like “The present work synthesised and evaluated second-generation polyamidoamine dendrimer-functionalized magnetic nanoparticles (Fe3O4@G2-PAD) for the detection of Pb2+ and Cd2+ ions in ambient fluids.” It is confusing to say “The present work…”. Some review papers are recommended for references: Coordination Chemistry Reviews 2022, 466, 214604; Journal of Bioresources and Bioproducts 2021, 6, 292-322; Nanoscale 2022, 14, 8216.
  • Thank you very much for the suggestions. However, the authors only cited related references in the manuscript. 
  1.     Some examples are not related to the topic of “natural”, please remove them.
  • The authors has revised and added information that related to the topic in the manuscript.
  1.     The conclusion part is too short. Some insightful conclusions and prospects should be offered.
  • The conclusions part has been revised, which reflects the findings.
  1. The authors should pay attention to the formation of references, especially the writing of superscripts and subscripts. 
  • The references have been formatted following the journal guidelines.

Round 2

Reviewer 2 Report

The topic of the review is good and attracting to readers. However, the content is not well organized and misleading. I do not think it is acceptable for Nanomaterials considering the good reputation of Nanomaterials. Comments are showed below.

1.     The logic of manuscript is confusing. How can the authors put “2.3. Other natural resources” together with “2.1. Carbon dots” and “2.2 Carbon nano-onions”? “2.3. Other natural resources” cannot be recognized as “Natural-derived nanomaterials”.

2.     The content is not well organized. The content should be organized by the detection mechanisms of nanomaterials or the issues influence the detection or the detection technologies other than by the type of biomass.

3.     “3. Natural resources for green synthesis of metallic nanoparticles” does not fall in the topic of this review.

4.     What is the relationship between “4. Engineered nanomaterials in detection of heavy metal ions” and “2. Natural-derived nanomaterials for detecting heavy metal ions”? These two sections are partly overlapped.

5.     Section 5 is missing. Otherwise, “6. Conclusion” should be “5. Conclusion”.

6.     The conclusion part is too routine. Insightful viewpoints should be given.

Author Response

Dear Reviewer,

The authors would like to inform you that the revised manuscript referred to above has corrected the suggestions and constructive comments of the reviewers and editors. The comments and suggestions highlighted were considered in the revised manuscript. The modifications, additions, and corrections appear in the article.

The topic of the review is good and attracting to readers. However, the content is not well organized and misleading. I do not think it is acceptable for Nanomaterials considering the good reputation of Nanomaterials. Comments are showed below.

  • Thank you for your constructive and insightful comments on this manuscript. The authors have revised the content of the manuscript accordingly.
  1. The logic of manuscript is confusing. How can the authors put “2.3. Other natural resources” together with “2.1. Carbon dots” and “2.2 Carbon nano-onions”? “2.3. Other natural resources” cannot be recognized as “Natural-derived nanomaterials”.
  • The authors have revised the subtopics and re-arrange the content of the manuscript.
  1. The content is not well organized. The content should be organized by the detection mechanisms of nanomaterials or the issues influence the detection or the detection technologies other than by the type of biomass.
  • The authors have organized the content accordingly.
  1. “3. Natural resources for green synthesis of metallic nanoparticles” does not fall in the topic of this review.
  • The authors have revised the subtopic accordingly.
  1. What is the relationship between “4. Engineered nanomaterials in detection of heavy metal ions” and “2. Natural-derived nanomaterials for detecting heavy metal ions”? These two sections are partly overlapped.
  • The section has been modified and organized based on the contents.
  1. Section 5 is missing. Otherwise, “6. Conclusion” should be “5. Conclusion”.
  • The section has been revised it accordingly.
  1. The conclusion part is too routine. Insightful viewpoints should be given.
  • The authors have rewritten the conclusion accordingly.